# Effect of Vitamin D on Graft-versus-Host Disease

**DOI:** 10.3390/biomedicines10050987

**Published:** 2022-04-24

**Authors:** Alfonso Rodríguez-Gil, Estrella Carrillo-Cruz, Cristina Marrero-Cepeda, Guillermo Rodríguez, José A. Pérez-Simón

**Affiliations:** 1Instituto de Biomedicina de Sevilla (IBIS/CISC), Universidad de Sevilla, 41013 Sevilla, Spain; arg@us.es; 2Department of Hematology, University Hospital Virgen del Roco, Instituto de Biomedicina de Sevilla (IBIS/CISC), 41013 Sevilla, Spain; estrellacarrillocruz@gmail.com (E.C.-C.); cristina.marrero.sspa@juntadeandalucia.es (C.M.-C.); grgarcia@gmail.com (G.R.)

**Keywords:** vitamin D, calcifediol, calcitriol, graft-versus-host disease, vitamin D receptor (VDR)

## Abstract

The different cell subsets of the immune system express the vitamin D receptor (VDR). Through the VDR, vitamin D exerts different functions that influence immune responses, as previously shown in different preclinical models. Based on this background, retrospective studies explored the impacts of vitamin D levels on the outcomes of patients undergoing allogeneic hematopoietic stem-cell transplantation, showing that vitamin D deficiency is related to an increased risk of complications, especially graft-versus-host disease. These results were confirmed in a prospective cohort trial, although further studies are required to confirm this data. In addition, the role of vitamin D on the treatment of hematologic malignancies was also explored. Considering this dual effect on both the immune systems and tumor cells of patients with hematologic malignancies, vitamin D might be useful in this setting to decrease both graft-versus-host disease and relapse rates.

## 1. Introduction

Despite its name, vitamin D is in fact a secosteroid hormone [1]. Currently, there is concern about vitamin D levels in the population at the worldwide level, with prevalences of severe vitamin D deficiency (defined as 25-OH-vitamin D serum levels lower than 30 nmol/L) ranging from 2.9% in the United States, 7.4% in Canada and 13% in Europe to more than 20% in India, Pakistan and Afghanistan [2,3,4,5]. Several studies addressed interventions in this deficiency in vitamin D, although clinical trials for its supplementation did not reach satisfactory results (reviewed in Amrein et al. [6]). The incidence of vitamin D deficiency is even higher among patients undergoing allogeneic hematopoietic stem-cell transplantation due to long-term hospitalizations or liver or renal toxicities, among other reasons.

## 2. Chemical Structure, Synthesis and Metabolism of Vitamin D

Vitamin D’s chemical structure, synthesis and metabolism were reviewed in [7]. In brief, it was initially discovered in 1919 by Edward Mellanby [8] as a micronutrient able to prevent rickets in dogs. Vitamin D is the common name assigned to a family of members, but usually refers to the precursor form vitamin D_3_ or cholecalciferol. Vitamin D_3_ is produced in the skin by a photolytic effect of UV light on 7-dehydro-cholesterol to produce previtamin D_3_ and by a subsequent thermal isomerization to produce vitamin D_3_. Vitamin D can also be obtained in the diet either as vitamin D_3_ from an animal origin or vitamin D_2_ (ergocalciferol) from vegetal and fungal origins. Vitamin D_3_ is further processed into 25-hydroxyvitamin D_3_ in the liver by the enzyme vitamin D_3_-25-hydroxilase, codified by the gene CYP2R1.

25-hydroxyvitamin D_3_ is the main circulating form and the marker clinically used to assess vitamin D_3_ levels. 25-hydroxyvitamin D_3_ is further processed into 1,25-dihydroxyvitamin D_3_, which is the active form [9,10,11], by the vitamin D-1α-hydroxylase, encoded by the gene CYP27B1 [12,13]. This step takes place mainly in the kidney, but many other tissues also express this gene, including several immune populations. Finally, vitamin D_3_ is deactivated by the enzyme vitamin D 24-hydroxylase, which is expressed in almost all cells [14], producing 24,25 dihydroxyvitamin D_3_, which is further processed and excreted through bile (Figure 1).

Vitamin D exerts its function mainly through binding to the vitamin D receptor (VDR), which belongs to the family of steroid nuclear receptors [15]. The VDR dimerizes with the retinoid X receptor (RXR) upon the vitamin D binding [16,17] and binds to DNA in the so-called vitamin D response element (VDRE) [18]. Interestingly, the VDR also has vitamin-D-independent actions [19]. This is the case of the roles of the VDR in hair follicle cycling [20] and in skin cancer development [21].

Several naturally occurring polymorphisms in the VDR gene were described [22,23,24,25,26] using restriction fragment-length polymorphisms (RFLP). Of special interest was the *Fok*I polymorphism, located in the second exon of the VDR mRNA. This polymorphism generates an alternative star codon that renders a protein three amino acids shorter (424 vs. 427 aas) with a higher transcriptional activity [26]. The *Bsm*I, *Apa*I and *Taq*I sites were also extensively studied. These three polymorphisms were mapped in the last intron of the gene, close to the 3′ UTR of the VDR mRNA, and are genetically linked. The VDR polymorphism has been associated with defects in bone metabolism (see [27] and additional references therein). The *Fok*I polymorphism has also been described to have an impact on the immune system [28].

## 3. Classical and Non-Classical Effects of Vitamin D

Beyond its classical effects on calcium and phosphate homeostasis and bone formation [1], vitamin D has also non-classical functions [29] in the regulation of hormone synthesis and secretion, including the parathyroid hormone (PHT), the fibroblast growth-factor 23 (FGF-23) or insulin in the cell proliferation in the skin, in cancer and in the immune system.

## 4. Effects on the Immune System

In the 1980s, it was described that vitamin D has multiple direct effects on the immune system’s function [30,31,32,33,34,35,36]. Before then, a first link between the immune system and vitamin D came from an observation that cod liver oil can be used for the treatment of tuberculosis [37,38]. Since then, many studies have elucidated the molecular mechanisms by which vitamin D affects immune cells. B and T lymphocytes, monocytes/macrophages, dendritic cells (DCs) and natural killer (NK) cells express the VDR [30,35,39,40], and most immune populations also express 1α-hydroxylase [39,41,42,43].

### 4.1. Effect on Innate Immune Cells

Monocytes and Macrophages: Both monocytes and macrophages express the VDR and 1α-hydroxylase [44], as aforementioned. In both cases, their expression is induced upon the stimulation of the toll-like receptors (TLR) 2/1 by pathogen-associated molecular patterns (PAMPs) [45] and interferon γ (IFN-γ) [41]. Vitamin D induces the expression of antimicrobial proteins, such as cathelicidin and β-defensin-2 [46,47], playing an important role in the first response to microbial infections. However, vitamin D skews the polarization of monocytes to a less pro-inflammatory phenotype, altering the cytokine secretion profile by changing MAPK1 signaling [48,49]. Additionally, vitamin D impairs the maturation of monocytes into dendritic cells [50] while favoring the phagocytic capacity of macrophages though the induction of complement receptors [51].

Dendritic cells: DCs form a complex system of different subsets that play a central role in the activation of the adaptive immune response through their antigen-presenting capacity to T cells [52]. The effect of vitamin D on DCs was reviewed by Bscheider and Butcher [53]. Vitamin D inhibits the differentiation, maturation, activation and survival of dendritic cells [54,55], which leads to a reduced activation of T cells. This tolerogenic state is driven by metabolic changes in the vitamin-D-treated DCs [56]. DCs activated in the presence of vitamin D also show altered trafficking properties [57]. Finally, DCs have been proposed to provide T cells with 25-hydroxy-vitain D3 in a paracrine fashion, inducing the expression of CCR10 and altering the migratory properties of these T cells [58].

Neutrophils: Neutrophils represent the major population of the innate immune compartment. Although they express the VDR and several genes modify their expression upon vitamin D treatments [59], the treatment’s effect on neutrophils’ functionality is controversial. Neutrophils exert their function using three different strategies: phagocytosis, degranulation and the formation of so-called neutrophil extracellular traps (NETs) [60]. NETs are web-like structures formed by proteins and DNA excreted by the neutrophils upon stimulation, which are able to trap, neutralize and kill bacteria but can also contribute to autoimmunity [61]. Vitamin D was described as preventing the endothelial damage induced by NETS in systemic lupus erythematosus (SLE) [62], but it was also shown to induce the formation of NETs within in vitro cultures [62].

NK cells: The effect of vitamin D on NK cells has not been exhaustively investigated. In vitro studies showed that vitamin D impairs NK’s differentiation from HSCs [63], favoring monocyte production. Mature NK cells, however, were not affected in cytotoxicity or IFN-γ secretion.

### 4.2. Effect on Adaptative Immunity

T lymphocytes: T cells express the VDR, and this expression is upregulated upon T cell activation [64]. Among T cells, Th1 and Th17 CD4 T cells show higher expression [65]. VDR knockout mice showed no significant changes in myeloid or lymphoid populations, but a reduced Th1 polarization with a downregulated IFN-γ secretion and increased IL4 production was observed upon stimulation [66]. CD4 and CD8αα T cells from VDR KO mice showed a reduced homing capacity to the gut due to reduced CCR9 expression levels [67]. Human T lymphocytes treated with vitamin D also showed reduced Th1 response [65,68]. TCR signaling is also affected by vitamin D. Phospholipase C γ-1(PLC γ-1) is a key signaling enzyme downstream of the TCR activation cascade, the expression of which is controlled by the VDR in human T cells [69]. In the presence of a vitamin D antagonist, the expression of PLC γ-1 is downregulated, and therefore, the TCR signal is impaired. Many studies showed the influence of vitamin D in Tregs (reviewed in [70]). Treatment with vitamin D induce immunotolerance by increasing Treg numbers in a DC-dependent manner [71,72] through the favoring of a tolerogenic phenotype of DCs. Vitamin D is also able to influence both IL10+ and Foxp3+ Tregs directly, promoting their expansion [73]. As mentioned previously, vitamin D enhanced VDR signaling through the upregulation of PLC γ-1. In Tregs, the activation of this axis leads to the expression of the anti-inflammatory cytokine TGF-β1 [74], increasing the regulatory properties of Tregs.

B cells: Similar to the case of T cells, B cells express low levels of the VDR in a resting state and upregulate it upon activation [75]. In vitro activated B cells showed decreased plasma cell differentiation and Ig secretion when they were cultured under vitamin D supplementation [31,75,76,77]. The targeting of the VDR with an agonist leads to the inhibition of B-cell-dependent allergic response in a murine model of type I allergies [78]. Additionally, vitamin D induces the production of IL10 up to threefold [79], suggesting a role in the development of regulatory B cells [80]. However, these effects have not been observed in vivo in human samples [81], and therefore, the actual role of vitamin D within B cells in vivo remains to be clarified.

Given the broad effects of vitamin D on immune cells, the consequences of vitamin D deficiencies on inflammatory and autoimmune diseases were extensively investigated (reviewed in Ao et al. and Hayes et al. [65,82]). In the past two years, the role of vitamin D in the immune response to COVID-19 also attracted great interest (reviewed in Ghelani et al. [83]). The importance of vitamin D in stem-cell transplantation will be discussed in the following section and was reviewed by Soto et al. [84] and Hong et al. [85]. Vitamin D’s impact on leukemia and hematopoiesis [86] and on cancer in general [87] were also recently reviewed.

The effects of vitamin D on immune cells are summarized in Figure 2.

### 4.3. Preclinical Models of Vitamin D in Immune Diseases and Solid Organ Transplantation

Several preclinical mouse models evaluating the impact of vitamin D on immune diseases were developed, including models evaluating solid organ transplants, experimental autoimmune encephalomyelitis (EAE), autoimmune diabetes, ulcerative colitis, systemic lupus erythematosus (SLE), autoimmune thyroiditis, collagen induced arthritis and graft-versus-host disease (GvHD).

Solid organ transplant models: Adorini et al. showed in 2003 that vitamin D, alone or in combination with mophetil mycophenolate, was able to prevent rejection in a heart-transplant model [88] through an increase in Treg numbers induced by tolerogenic DCs. More recently, Xi et al. used a combination of anti-CD40L antibodies and vitamin D to prevent memory T-cell -mediated rejections in heart transplants [89]. The use of vitamin D in mouse models of pancreatic islet transplantation was reviewed by Infante et al. [90].

Autoimmune diabetes: Autoimmune diabetes mouse models based on the non-obese diabetes (NOD) strain were also used to study the role of vitamin D in diabetes’ development in addition to its role in islet transplantation. Vitamin D reduces the immune response to pancreatic islands by increasing Tregs [91] and lowering the pro-inflammatory cytokine production [92]. Interestingly, VDR knockout NOD mice presented an unaltered presentation of diabetes compared to VDR+/+ mice [93].

Experimental autoimmune encephalomyelitis: EAE is a preclinical model for multiple sclerosis. Vitamin D was shown to reduce EAE in an IL10-signaling-dependent manner [94] by altering chemokine secretion and monocyte trafficking [95] Rag1-dependent cells are essential for this response [96]; however, CD8+ cells are not necessary [97]. The conditional deletion of the VDR in T cells abolished the beneficial effect of vitamin D on EAE [98].

Systemic lupus erythematosus: The mouse strain MLR/1 is a model of spontaneous SLE syndrome. The treatment of this strain with vitamin D reduces the appearance of some manifestations of the disease [99]. In another model of SLE, pristine-induced lupus [100], vitamin D alleviates arthritis but does not reduce renal injuries [101].

Ulcerative colitis: Two widely used mouse models of ulcerative colitis are IL10 KO mice and dextran sodium sulfate (DSS)-induced colitis. In the first case, ulcerative colitis is developed spontaneously in a TNF-α-signaling-dependent manner. The severity is lower when high calcium or 1–25 di-hydroxy-vitamin D3 is included in the diet, while IL10 VDR double-KO mice develop a fulminating form of the disease [102]. In the case of DSS-induced colitis, the deletion of the VDR renders a hypersensitivity to the agent [103] and a vitamin D deficiency leads to an impaired antimicrobial gut response and increased colitis predisposition [104].

Stem-cell transplantation and GvHD: Despite the abundance of animal models of GvHD, to date, the published reports on the effects of vitamin D on GvHD animal models are scarce. In 2001, Pakkala et al. reported that the vitamin D analog MC1288 prevented acute GvHD in rats [105]. More recently, Taylor et al. noted that vitamin D can alleviate GvHD in allogeneic hematopoietic stem-cell transplantation recipients. Using VDR KO donors, the effect was retained, indicating that the vitamin D effect was recipient and not donor dependent [106].

## 5. Vitamin D in the Clinical Setting

### Vitamin D Compounds Available in the Clinical Setting

The natural compounds ergocalciferol (vitamin D_2_), cholecalciferol (vitamin D_3_), calcifediol (25-hydroxyvitamin D_3_) and calcitriol (1,25-dihydroxyvitamin D_3_) are available for use in clinics as supplements. Other synthetic products can be employed as so-called analogs.

A meta-analysis of randomized controlled trials that directly compared the effects of ergocalciferol and cholecalciferol confirmed that cholecalciferol increases serum 25-hydroxyvitamin D faster than ergocalciferol; this may be due to an affinity for the VDR [107]. Cholecalciferol and calcifediol are commonly administered for vitamin D deficiencies, although calcifediol is faster in action, more potent and has a shorter half-life compared to the pro-hormones [108].

Chronic kidney disease (CKD) generates hyperparathyroidism, osteomalacia and adynamic bone disease. In CKD patients, calcifediol normalizes vitamin D levels and decreases high PTH concentrations.

Calcitriol is preferably used in cases of secondary hyperparathyroidism in patients with CKD and in patients with hypocalcemia and normal renal function as it increases the intestinal calcium absorption [109]. In patients with CKD, the use of calcitriol has a risk of hypercalcemia and vascular calcification.

In this context, several synthetic vitamin D analogs can be used: paricalcitol (1,25 dihydroergocalciferol), doxercalciferol (1-alpha-ergocalciferol), alfacalcidol (1-alpha-hydroxyvitamin D_3_) and maxacalcitol (22-oxacalcitriol) [110]. All of them can be used in the treatment of secondary hyperparathyroidism in CKD patients, although paricalcitol and alfacalcidol might be related to a lower risk of hypercalcemia and hypophosphatemia [111].

The recommended doses depend on whether or not a subject has a vitamin D deficiency. The diagnosis of a vitamin D deficiency is established by noting low serum concentrations of 25-hydroxyvitamin D. The reference values are controversial and differ between populations due to diet intake, age, geography, sun exposure, etc. The Institute of Medicine (IOM) committee [112] proposed a reference value for healthy populations of above 20 ng/ mL in serum, while the International Osteoporosis Foundation (IOF) defined it as above 30 ng/ mL [113]. A vitamin D deficiency staging has been proposed [114,115] in which vitamin D insufficiency is defined when serum 25-hydroxyvitamin D levels are below 50 nmol/liter (20 ng/mL). This is associated with mild elevations of serum iPTH and biochemical markers of bone turnover. A moderate vitamin D deficiency (25-hydroxyvitamin D serum levels below 25 nmol/liter or 10 ng/mL) is associated with a moderately increased serum iPTH concentration and a high bone turnover. In a severe vitamin D deficiency (serum 25-hydroxyvitamin D levels lower than 12.5 nmol/liter or 5 ng/mL), patients may be at risk of rickets and/or osteomalacia. Moreover, a maximum reference value of 60–70 ng/ mL was proposed [116].

In the healthy population, the recommended doses of cholecalciferol are 400 international units (IU)/day for infants (1 IU is equal to 0.025 mcg), 600 IU/day for children and adults until the age of 70 (including pregnant and lactating women) and 800 IU/day above this age [112]. The American Geriatrics Society (AGS) and the National Osteoporosis Foundation (NOF) recommend 800 UI to 1000 UI daily to reduce the risk of fractures and falls in people ≥65 years.

In patients with vitamin D deficiencies, higher doses are needed. To find the proper dose, a deficiency calculation should be considered. For every 100 units (2.5 mcg) of added vitamin D3, serum 25-hydroxyvitamin D concentrations will increase by 0.7 to 1.0 ng/mL (1.75 to 2.5 nmol/L) [117]. In cases of severe deficiencies, 4000 to 6000 IU daily should be given for the first 4–6 weeks, followed by a dose adjustment in accordance with the biochemical response monitored at 3-month intervals to achieve the recommended maintenance dose and then continue monitoring at 6-month intervals [118]. Different dosage modalities were tested with overlapping results. Therefore, vitamin D can be prescribed daily, once a week or once a month as it has a half-life of 2–3 weeks and is released slowly from storage in the fat [119]. In this regard, obesity has a potential impact on the efficacy of vitamin D treatment, however, to our knowledge, this impact has not been specifically demonstrated in the available studies, and probably it would be resolved with stoss doses and vitamin D level monitoring.

## 6. Vitamin D and Hematologic Malignancies

The potential antitumor effects of vitamin D and low serum levels of 25-hydroxyvitamin D were reported in many neoplasms, leading researchers to consider a potential role of vitamin D in the treatment and prevention of cancer. Nevertheless, in a randomized placebo-controlled trial carried out in more than 25,000 subjects, supplementation with cholecalciferol at 2000 UI daily did not result in lower incidences of invasive cancer than a placebo [120].

The ability of vitamin D to promote differentiation and apoptosis was demonstrated during in vitro and preclinical studies on myelodysplastic syndromes (MDS) and acute myeloid leukemia (AML). Some degree of responses was observed with vitamin D in these neoplasms, although the evidence was not strong enough to set recommendations in the clinical setting. Therefore, the use of vitamin D and analogs are under continuous investigation.

Calcitriol was discovered in 1981 to induce the monocytic differentiation of the human promyelocytic leukemia cell line HL-60 [121]. Later, a similar effect was observed in other cell population lines, such as THP-1 (monoblasts), HEL (bipotent erythroblasts/monoblasts) and M1 (late myeloblasts) [122]. The use fo all-trans retinoic acid (ATRA) and calcitriol for the treatment of acute promyelocytic leukemia was proven to be an effective synergistic combination therapy for inducing differentiation and impairing cell growth. [123,124]. In addition, the analog KH1060 (modified 20-epi-1,25 dihydroxyvitamin D_3_) in combination with ATRA was proven to be synergic; they induce differentiation, proliferation inhibition and the induction of apoptosis [125,126]. Other analogs were tested in leukemic cells and shown to be more potent in vitro than calcitriol [126,127]. Moreover, a combination of paricalcitol and arsenic trioxide potently decreased the growth and induced the differentiation and apoptosis of AML cells. [128].

Low serum levels of vitamin D were reported in MDS [129], although their association with prognosis remains controversial. In a study by Pardanani, vitamin D levels did not correlate with prognosis in a series of 409 patients diagnosed with different myeloid neoplasms and MDS [130]. By contrast, levels of vitamin D were an independent predictor of survival in a retrospective study of 58 patients with MDS or secondary oligoblastic AML treated with 5-azacytidine (AZA) with an estimated probability of a 2 year overall survival of 40% for a high-level vitamin D group versus 14% for a low-level vitamin D group (*p* < 0.05). The AZA and 25-hydroxyvitamin D_3_ combination was also tested in vitro, showing a synergistic effect [131]. Similarly, a worse relapse-free survival was observed in AML patients with low vitamin D levels [132].

Regarding clinical studies on myelodysplastic syndromes, Koeffler [133] reported a minor response in 8 out of 18 patients treated with calcitriol at doses > 2 μg/day, but hypercalcemia was also observed in eight patients. Mellibovsky [134] reported responses in 11 out of 19 patients treated with calcitriol (0.25–0.75 μg/day) or calcifediol (266 μg three times a week). No cases of hypercalcemia were registered. In this study, no correlation was observed between the baseline levels of vitamin D metabolites and response.

Furthermore, two trials with paricalcitol at a dose of 8 μg/day [135] and doxercalciferol at a dose of 12.5 μg/day [136] did not show a clinical benefit. By contrast, there is evidence of a potential effect on progressions to AML. In this regard, Motomura et al. randomized a series of 30 patients to receive alfacalcidol versus a supportive treatment [137]. Only one of the 15 patients who received alfacalcidol progressed to AML versus seven in the control group. Alfacalcidol also demonstrated an ORR of 30% when combined with menatetrenone [138]. In addition, a study of 63 patients with myelodysplastic syndromes (MDS) and 15 patients with acute myelogenous leukemia (AML) were randomized to receive low doses of ara-C or low doses of ara-C in combination with 13-cis-retinoic acid (13-CRA) and 1 alpha-hydroxyvitamin D_3_, showing that the addition of 13-CRA and 1 alpha-hydroxyvitamin D_3_ had no impact on survival or remission rates. However, a trend toward a lower rate of progression from MDSs to AML was found (*p* = 0.0527) [139]. Moreover, erythroid responses as high as 60% were reported in MDS patients with low-risk International Prognostic Scoring System (IPSS) scores treated with a combination of EPO, 13-CRA and calcitriol and with a median response duration of 16 months [140].

In 29 elderly patients with AML, a combination of cytarabine (20 mg/m^2^/day for 21 days), oral hydroxyurea (500 mg twice a day) and calcitriol (0.5 μg twice a day) followed by a calcitriol maintenance was tested, achieving a 79% overall responses (34% partial and 45% complete remission) with a duration of 9.8 months. Two cases of hypercalcemia were observed [141].

There are also data on the antitumoral effect of vitamin D on lymphoid neoplasms. A significant association was described between low serum vitamin D levels and survival in patients diagnosed with follicular lymphoma [142]. Patients included in SWOG clinical trials who were vitamin D deficient (<20 ng/mL; 15% of the cohort) had adjusted PFSs and overall survival hazard ratios of 1.97 (95% CI of 1.10 to 3.53) and 4.16 (95% CI of 1.66 to 10.44), respectively (a median follow-up of 5.4 years) [143].

Furthermore, a prospective study performed on 983 patients with non-Hodgkin’s lymphoma showed that vitamin D insufficiencies (<25 ng/mL) in DLBCL were associated with an inferior EFS (hazard ratio (HR) of 1.41; 95% CI of 0.98 to 2.04) and OS (HR of 1.99; 95% CI of 1.27 to 3.13). T-cell lymphoma patients also had inferior EFSs (HR of 1.94; 95% CI of 1.04 to 3.61) and OS (HR of 2.38; 95% CI of 1.04 to 5.41) [144].

A meta-analysis investigated the association between various measures of vitamin D statuses and the risk of developing non-Hodgkin’s lymphoma (NHL). Significant protective effects of overall sunlight/UVR exposure on NHL were observed, although the risk estimates were inconsistent when dietary vitamin D intakes and vitamin D levels were measured [145]. In mantle cell lymphoma, vitamin D deficiencies were independent prognosis factors for PFS (hazard ratio (HR) of 3.713; 95% confidence interval (CI) of 1.822–7.565; *p* < 0.001) and OS (HR of 8.305; 95% CI of 2.060–33.481; *p* = 0.003). This was confirmed in a multivariate analysis in which mantle cell’s international prognostic index was included [146]. Similarly, decreases in PFS (HR of 3.323; 95% CI of 1.527–7.229; *p* = 0.002) and OS (HR of 5.819; 95% CI of 1.322–25.622; *p* = 0.020) were observed in patients with Hodgkin’s lymphoma [147]. Another study that supported a poor prognosis among vitamin-D-deficient patients with Hodgkin’s lymphoma was carried out on 351 patients included in German Hodgkin’s Study Group clinical trials (HD7, HD8 and HD9). Interestingly, there was evidence of an improved outcome in patients with DLBCL administered rituximab-based treatments who previously were deficient/insufficient in vitamin D and achieved normal levels after vitamin D3 supplementations [148]. A protective effect of vitamin D supplementation against the development of lymphoid malignancies was reported in a randomized controlled trial that recruited 34,763 women and aimed to evaluate their incidences of skeletal fractures and cancer. Women receiving vitamin D and calcium had HRs of 0.77 (95% CI of 0.59–1.01) and 0.46 (95% CI of 0.24–0.89), respectively, for cancer incidences and mortality. Despite some limitations, these results provided support for the design of vitamin D clinical trials [149]. Several clinical trials are ongoing to address the impacts of vitamin D replacement on the prognosis of lymphoid malignancies (Table 1). A vitamin D replacement strategy in vitamin-D-insufficient patients with lymphoma or chronic lymphocytic leukemia was successfully performed by Sfeir et al. [150]. Target vitamin D levels of ≥30 ng/mL were achieved in 97% of the patients at the end of a 12-week induction period. This strategy is being now evaluated in a clinical trial (NCT01787409) to analyze vitamin D’s impact on the prognoses.

Regarding multiple myeloma, preclinical studies showed the activity of the vitamin D analogue EB1089 in the cell line H929. This agent promotes apoptosis and induces a cell-cycle arrest through the downregulation of cyclin-dependent kinases [152,153]. Although a vitamin D deficiency is common in multiple myeloma, supplementation was not found to improve the outcomes of patients. Currently, the recommendations of vitamin D supplements are to improve bone and immune health in MGUS and MM patients [154].

## 7. Vitamin D and Allogeneic Stem-Cell Transplantation: Effect on Graft-Versus-Host Disease (GvHD)

Patients undergoing allogenic hematopoietic stem-cell transplantation (allo-HSCT) have a higher risk of a vitamin D deficiency than the healthy population due to multiple factors, as previous studies demonstrated [155]. Long-term hospitalizations decrease patients’ sun exposure, and such patients are even counseled to minimize their unprotected exposure to sunlight due to an increased risk of non-melanoma skin cancer as well as a potential activation of chronic GvHD [156]. Moreover, Vitamin D’s absorption by the small intestine is often decreased due to gastrointestinal GvHD, infectious colitis or mucositis. Toxic treatments used in allo-HSCT also play a role in this deficiency; they can affect absorption too, reducing oral intakes due to gastrointestinal toxicity, and can interact with calcitriol through CYP3A4 (e.g., calcineurin inhibitors, which, similar to vitamin D, are substrates of this cytochrome). Finally, other possible complications usually lead to renal or hepatic dysfunctions affecting vitamin D statuses as well [85,156].

GvHD is one of the most frequent and severe complications after allo-HSCT; it is caused by the cytotoxic effect of the donor T lymphocytes on the recipient organs. Acute GvHD physiopathology involves T lymphocytes, natural killer cells, and the innate immune system [157]. In the case of chronic GvHD, a complex interaction between B and T lymphocytes leads to the production of auto-antibodies, cytokines and chemokines, which in turn induce the activation of the monocytic macrophage system. Growth factors, such as TGF-β, produced by wound-healing macrophages induce a fibroblast proliferation and a subsequent fibrosis of target organs [158,159,160].

Considering the previously mentioned influence of vitamin D on the regulation of the immune response and its potential effect on several hematologic malignancies, its role on allo-HSCT has been a great focus of interest.

### 7.1. Vitamin D Levels: Impacts on Allogenic HSCT Outcomes

Patients undergoing hematopoietic stem-cell transplants (HSCT) are at high risk for vitamin D deficiencies before and after their transplants [161,162]. The prevalence of vitamin D deficiency was reported to be approximately 30% in the general population and is significantly higher in this setting of HSCT (70% before transplants and 90% after transplants) [162,163]. Since vitamin D levels are not always monitored in HSCT patients and there is a high prevalence of vitamin D deficiencies among them, Kenny et al. established a workflow for monitoring and treating vitamin D deficiencies and determining whether or not therapeutic vitamin D levels can be achieved post-transplant using an HSCT-specific vitamin D algorithm [156]. The initial replacement doses were serum vitamin-D-dependent, and again, a dose adjustment-based level was measured at several points. With the implementation of this algorithm, vitamin D deficiencies decreased from 72.9% pre-transplant to 26.4% post-transplant. Vitamin D supplementations in HSCT patients do not always achieve optimal serum vitamin D levels [164]. Therefore, a more intensive vitamin D replacement than recommended for the general population may be required in HSCT patients [156].

Several studies described a link between GvHD’s incidence and/or severity and vitamin D deficiency [165,166,167,168,169]. Some of them [166,168,169] specifically described an impact of vitamin D deficiency on chronic GvHD incidences. By contrast, other studies did not find a significant correlation [162,170,171,172,173,174,175,176,177,178].

When we look at survival rates, it is difficult to make a definitive statement. Beebe et al. and Hansson et al. [168,171] described a worse overall survival rate, and Perera et al. [176] described a higher mortality rate among patients with vitamin D deficiencies, while Bhandari et al. [175] found that vitamin D levels correlate with overall survivals upon considering vitamin D levels in the follow-up, but not just vitamin D levels before HSCTs.

In the largest study by Radujkovic et al. [177] on 492 patients, a significant association between vitamin D deficiency and an inferior overall survival rate was described (Hazard ratio of 1.78; *p* = 0.007). This effect was due to a higher risk of relapse (HR of 1.96; *p* = 0.006) in myeloid diseases. This study did not find a relationship between vitamin D levels and incidences of acute or chronic GvHD.

In a meta-analysis, Ito et al. [179] observed that lower vitamin D levels were associated with a significantly poorer overall survival rate (HR of 1.50; 95% CI of 1.03–2.18) and a higher relapse rate (HR of 2.12; 95% CI of 1.41–3.19), while no significant impact on non-relapse mortality (NRM) was described (HR of 1.23; 95% CI of 0.72–2.10).

Another meta-analysis [180] concluded that vitamin D deficiency was not significantly associated with a higher risk of GvHD, although there was a trend for both acute (HR of 1.06; 95% CI of 0.74–1.53; *p* > 0.05) and chronic (HR of 1.75; 95% CI of 0.72–4.26; *p* > 0.05) GvHD. All these results are summarized in Table 2. With this background, Hong et al. proposed that vitamin D levels should be monitored in all patients prior to allo HSCT and every 3 months thereafter.

For monitoring purposes, as previously mentioned, the main circulating metabolite of vitamin D in serum is 25-hydroxyvitamin D, and it is considered the most reliable marker [182]. However, Peter et al. [178] noted the underestimated role of 1,25-dihydroxyvitamin-D_3_ and its value in predicting outcomes after allo-HSCTs. They measured 1,25-dihydroxyvitamin-D_3_ in 143 patients and compared their findings with 25-hydroxyvitamin D levels and found that only peritransplant 1,25-dihydroxyvitamin-D_3_ deficiencies were significantly associated with higher 1-year NRM. Afterwards, they studied 365 additional patients and again showed that patients with 1,25-dihydroxyvitamin-D_3_ levels below 139.5 pM had a 3.3-fold increased risk of NRM (Cox-model unadjusted *p* < 0.0005; adjusted *p* = 0.001).

In addition to the differences found among the studies, there are several factors that could make even harder to reach definitive conclusions about the relationship between vitamin D levels and HSCT outcomes, such as the underlying malignancy or the patient’s age. For example, Duncan et al. [183] pointed out that older age at enrollment was a risk factor for vitamin D deficiency in a multivariable analysis and Hansson et al. [168] described that patients with low levels of vitamin D were older than those with sufficient levels (*p* = 0.025). These results have not been confirmed by other authors [156,165,177]. If we focus on specific outcomes, Glotzbecker et al. [166] highlighted that low pretransplant vitamin D level remained a significant independent factor associated with the risk of cGvHD when adjusted for patient age in a multivariable competing risk model. Similar data were reportd by Peter et al. [178]. 

Other confounding factors that we likely need to be aware of, especially when studying the risk of GvHD, are the HSCT characteristics, such as stem cell source and transplant platforms (related donor or not, haploidentical transplantation).

### 7.2. Studies Evaluating the Efficacy of Vitamin D Administration

With these data in mind, several studies evaluated the potential benefits of the administration of different subtypes or doses of vitamin D in the allo-HSCT setting (Table 3). Kenny et al. [156] proposed an ergocalciferol (or cholecalciferol) dose of up to 50,000 IU orally once weekly, and they found only 19.7% allogenic-deficient patients after the transplants (69.7% were deficient before the transplants). They concluded that aggressive post-transplant vitamin D repletions decreased the incidence of vitamin D deficiency.

Other studies, included in Table 3, related vitamin D supplementations with HSCT outcomes. One of them [185] analyzed the impact of vitamin D administrations on patients with active chronic GvHD, finding an improvement in severity and remarkable reductions in relapses and progressions.

Bhandari et al. [186] designed a study in a pediatric population to evaluate whether a single weight-based ultra-high dose of vitamin D—or Stoss dose—was more effective than a standard supplementation to achieve a pre-HSCT vitamin D sufficiency and reduce the incidence of HSCT-related complications that are associated with immune-mediated endothelial damage [186]. Stoss doses were given to 33 patients 14 days before conditioning, and then routine maintenance supplementation was given before day 100 in case of insufficiencies. The outcomes were compared to a historical cohort of 136 patients treated with standard supplementation. Low levels of vitamin D were present in 61% of the patients, and 97% of them maintained a vitamin D sufficiency after the Stoss doses compared to 67% (*n* = 10/15) of the patients in the historical control who were on standard supplementation at the time their total 25(OH)D levels were assessed (*p* = 0.013). There was a trend toward a lower combined incidence of HSCT-related complications in patients receiving Stoss-dosed vitamin D, unlike in the historical control (25% (*n* = 7/28) versus 42% (*n* = 57/136); *p* = 0.055). A randomized phase-four trial was performed to assess the safety and efficacy of Stoss doses versus standard vitamin D replacements; the study is awaiting results (NCT03176849). A summary of ongoing trials of vitamin D in HSCT settings is shown in Table 4.

The Alovita trial was a prospective study we conducted in which 150 patients older than 18 years from seven Spanish centers were included from May 2011 to February 2014 [188,189]. Three consecutive cohorts with 50 patients each were included: a control group (CG) that did not receive cholecalciferol (vitamin D_3_), a second cohort or low-dose group (LdD) that received 1000 IU of vitamin D_3_ per day and a high-dose group (HdD) that received 5000 IU of vitamin D_3_ per day. Vitamin D_3_ was given orally from day −5 before transplants until day +100 after transplantations.

Regarding toxicity, no serious adverse events (specifically, no cases of hypercalcemia) were reported.

Vitamin D_3_ supplementation was proved to be effective in terms of reductions in chronic GvHD incidences. A decrease of both overall as well as moderate plus severe cGvHD incidences at 1 year were observed in LdD (37.5% (95% CI of 24.9–56.4) and 19.5% (95% CI of 10.4–36.7), respectively) and in HdD (42.4% (95% CI of 29.3–61.4) and 27% (95% CI of 16.1–45.2), respectively) compared with patients who did not receive vitamin D (67.5% (95% CI of 54.1–84.3) and 44.7% (95% CI of 31.2–64.2), respectively; *p* = 0.019 for the overall incidence and *p* = 0.026 for the moderate plus severe cGvHD incidences, respectively). No significant differences were observed in terms of cumulative overall and grades 2–4 acute GvHD incidences or cumulative incidence of relapse at 1 year. No significant differences in neither disease free survival (DFS) nor OS were observed with a median follow-up of 2 years

This effect correlated with several biological parameters. The most significant differences between the three cohorts were decreases in both the percentages and absolute numbers of circulating B cells on day 100 for the LdD and HdD subgroups compared with the CG subgroup; a markedly modified ratio of naïve/memory/effector T cells with lower numbers of circulating naïve CD8+ among the patients receiving vitamin D compared with those who did not receive it and significantly lower expressions of CD40L as activation marker among the patients receiving vitamin D. These findings are concordant with an increase in immune tolerance development at the same time as the survival and expansion of naïve T and B donor cells are impaired.

Next, we performed a retrospective study among the patients previously included in the Alovita trial to identify which factors might influence the effects of vitamin D on cGvHD; in particular, we focused on the evaluation of the different VDR SNPs among patients and their respective donors who had genomic DNA stored before transplants [189]. The patients were divided into two groups: a vitamin D group that received 1000 or 5000 UI daily (*n* = 71) and a control group (*n* = 36). We investigated the SNPs *Fok*I (rs2228570 T/C), *Bsm*I (rs1544410 A/G), *Apa*I (rs7975232 C/A) and *Taq*I (rs731236 T/C) in 107 patients and 102 donors. We found that the *Bsm*I, *Apa*I and *Taq*I alleles were in strong disequilibrium. In contrast, *Fok*I did not demonstrate any association with *Bsm*I, *Apa*I or *Taq*I. Overall, there were no significant differences in the incidences of cGvHD depending on the patients’ or donors’ SNPs. In contrast, the VDR genotypes significantly influenced the impacts of vitamin D administration on cGvHD incidences. The administration of vitamin D significantly influenced the risk of overall cGvHD among patients with *Fok*I CT (the cGvHD incidences were 22.5% (95% CI of 8.8–39) vs. 80% (95% CI of 30.8–95) for the patients receiving and not receiving vitamin D, respectively; *p* = 0.0004). The same genotype also influenced the risk of moderate to severe cGvHD. We also evaluated the benefits obtained from the post-transplant administration of vitamin D depending on the most frequent patients’ *Bsm*I/*Apa*I/*Taq*I haplotypes. In this regard, the patients carrying the GGT/GGT genotype had the greatest benefit from receiving vitamin D in terms of cGvHD incidence, although we could not confirm that data in a multivariate analysis. In that analysis, a significant interaction regarding the risk of overall cGvHD was observed between the *Fok*I genotype and vitamin D administration. Accordingly, the risk of cGvHD in the patients treated with vitamin D was lower among the patients carrying the *Fok*I CT genotype (adjusted hazard ratio (aHR) of 0.143; 95% CI of 0.045–0.452; P interaction < 0.001). In addition, we performed an analysis to evaluate the vitamin D supplementation’s impact on the survival rate, relapse incidence and non-relapse mortality rate without finding any association.

Emphasizing the finding of a decreased risk of cGvHD among a specific SNP (*Fok*I) of the recipients, the effects of vitamin D on dendritic cell populations might be the most relevant to justify the impact of vitamin D on cGvHD incidences. Some subtypes of dendritic cells from hosts persist after engraftments; therefore, vitamin D binding to the VDR would inhibit the cells’ differentiation and maturation and would decrease alloreactive T-cell activation at the same time as it would upregulate the tolerogenic properties selectively in myeloid dendritic cells.

In summary, the effects of vitamin D on hematopoietic cells, especially on the different cell subsets from the immune system, together with the previously mentioned data and the excellent toxicity profile support its use in the allo-HSCT setting in an attempt to decrease cGvHD. Additional studies are required to further explore its efficacy for GvHD prophylaxis which, along with the proper dose, time of administration and monitoring strategies should also take into consideration patients and transplant characteristics.

## Figures and Tables

**Figure 1 biomedicines-10-00987-f001:**
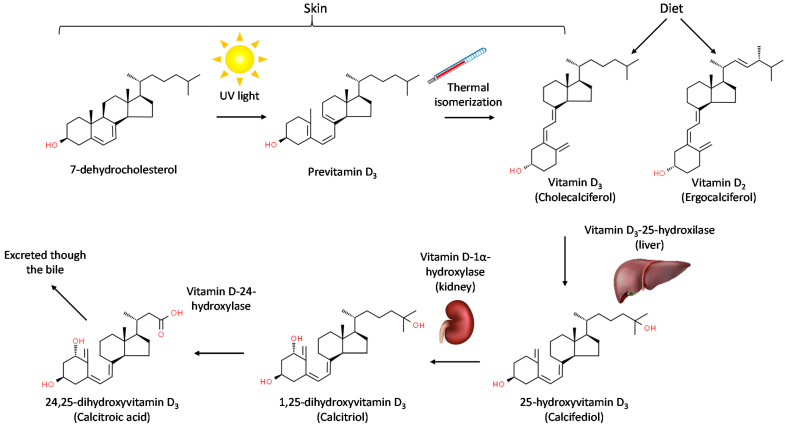
Metabolism of vitamin D.

**Figure 2 biomedicines-10-00987-f002:**
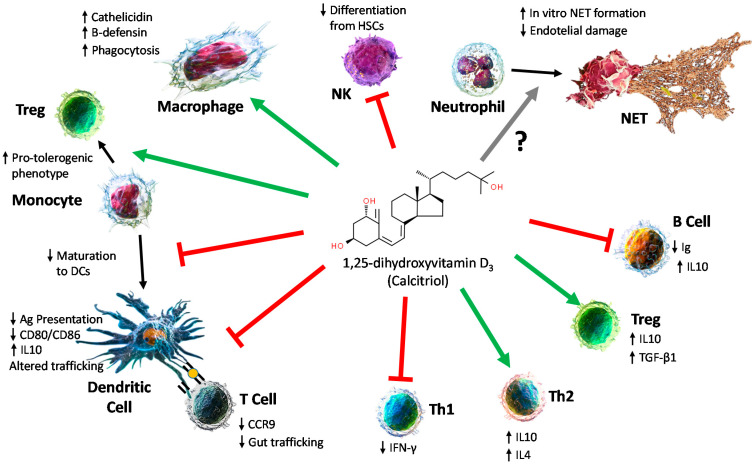
Summary of vitamin D’s effects on immune cells.

**Table 1 biomedicines-10-00987-t001:** Interventional studies evaluating the efficacy of vitamin D’s administration in treating hematologic malignancies.

	Study	*N*	Intervention	Vit. D Levels Median/Range	Endpoints
**MDSs** **Koeffler et al.,** **1985 [133]**	NR	18	Calcitriol (>2 mcg)	---	MR and PR 44% (8/18)
**MDSs** **Motomura et al., 1991 [137]**	Phase II	30	Alfacalcidol (4–6 mcg/day) vs. no therapy	---	Progression to AML Alfacalcidol: 6% (1/15) No therapy: 46.6% (7/15)
**MDS** **low and high IPSSs** **Koeffler et al.,** **2005 [135]**	NR	12	Paricalcitol (8 μg/day and increments of 8 μg/day every 2 weeks)	---	OR: 0%; 1/12 patients’ platelet counts achieved normal range for 5 weeks
**MDSs and CMML** **low-int1 IPSSs** **Mellibovsky et al., 2001 [134]**	NR	19	Calcifediol (266 mcg 3 times a week; *n* = 5) or calcitriol (0.25–0.75 mcg/d; *n* = 14)	Increased from 9.4 ± 4.6 ng/ mL to 37.5 ± 44.2 (*p* = 0.003)	OR: 57% (11/19); No hypercalcemia
**MDSs and CMML** **Petrich et al.,** **2008 [136]**	Phase II	15	Doxercalciferol (12.5 mcg/day for 12 weeks)	---	No responses
**MDSs with low IPSSs and int-1** **Akiyama et al.,** **2010 [138]**	Phase II	20	Alfacalcidol (0.75 mcg/day) + menatetrenone (45 mg for 1 year if response)	---	ORR: 30% (6/20)
**MDSs and AML** **Hellström et al., 2009 [139]**	Phase III	63 MDS 15 AML	Arm 1 Ld ara-C vs. arm 2 Ld ara-C + 13-CRA and alfacalcidol	---	Similar OS, ORR and DOR; progressed from MDSs to AML: 44% vs. 20% (*p* = 0.0527)
**MDSs** **low-int-2 IPSSs** **Ferrero et al.,** **2008 [140]**	Phase II	63	EPO + 13-CRA + calcitriol	---	RAEB1 OS 14 months; non-RAEB1 OS 55 months; erythroid response 60% (93% in low-risk patients)
**MDSs and CMML** **Siitonen et al., 2007 [151]**	NR	19	Valproic acid (dose adjusted by levels) + 13-CRA (10 mg/12 h) + calcitriol (1 mcg)	---	Blood improvement: 3/19 patients (16%); 8/19 discontinued (side effects, no hypercalcemia)
**AML (elderly)** **Slapak et al.,** **1992 [141]**	NR	29	- Ld Ara-C + hydroxyurea + calcitriol (0.5 μg/12 h)	---	ORR: 79%; CR: 45%/PR: 34%; DOR: 9.8 months
**MDSs—IPSSs 0/I** **NCT00068276** **Ongoing**	Phase II	36	Cholecalciferol (doses not specified)		Safety and efficacy
**CLL** **NCT01518959** **Ongoing**	Phase III	31	Cholecalciferol (180.000 IU monthly) vs. placebo		5 years OS, PFS and TTF 5-year lymphocyte count
**Aggressive NHL** **Hohaus et al.,** **2018 [148]**	NR	155	Cholecalciferol loading phase (25,000 IU daily) and maintenance phase (25,000 IU weekly)	Vitamin D pre-treatment 14 ± 1.4 ng/mL and post-treatment 33 ± 1.4 (*n* = 81; *p* < 0.0001)	Independent prognostic parameters for EFS 25(OH)D levels < 20 ng/mL; HR of 2.88; *p* < 0.02 IPI HR of 2.97; *p* < 0.002 No hypercalcemia
**NHL and CLL** **Sfeir et al.,** **2017 [150]** **NCT01787409**	Phase I/II	158	Cholecalciferol (50.000 IU weekly for 12 weeks; if <30 ng/mL: 50.000 IU twice weekly; when ≥30 ng/mL: 50.000 IU/month)	Vitamin D deficiency 45% (*n* = 71); mean ± SEM 17 ± 5 ng/mL	97% vitamin D insufficient group reached ≥30 ng/mL prior to follow-up period of 3 years, during which these levels were maintained
**NHL and CLL** **NCT01787409** **Ongoing**	Phase I/II	713	Cholecalciferol PO (once weekly for 12 weeks and then once monthly for a total of 36 months)	---	12 months EFS; 36 months treatment free; 5 years ORR and OS; 5 years TTF (CLL patients)
**NHL and CLL** **NCT02553447** **Ongoing**	Early phase I	370	- Arm I: high-dose cholecalciferol PO daily - Arm II: low-dose cholecalciferol PO daily - Arm III (control)	---	3 years PFS; 3 years OS
**Indolent NHL (ILyAD clinical trial)** **NCT03078855** **Ongoing**	Phase III	210	Weekly rituximab (4 weeks + - Arm 1: Cholecalciferol (2.000 IU daily) or - Arm 2: placebo	---	3 years PFS and OS; response to rituximab (reduction of lymphoma burden by at least 50%)
**Diffuse large B cell lymphoma—65 years and older** **(FIL_PREVID)** **NCT04442412** **Ongoing**	Phase III	430	- Arm A: 7 days of prephase oral prednisone - Arm B: 7 days of prephase oral prednisone and cholecalciferol (25.000 IU/day), then 25.000 IU/week - Both followed by six courses of R-CHOP R-miniCHOP/21 days	---	54 months PFS, OS and EFS; 54 months RR and EDR; 54-month rate of ECOG changed after prephase; rate of patients with 25(OH)D levels corrected at Cycle 2; time-to-deterioration physical functioning and fatigue at Cycle 2
**Untreated early-stage CLL (or SLL)** **Ongoing**	Phase II	35	Curcumin + oral daily cholecalciferol (on days 1–28 for six cycles; if PR, treatment up to 2 years)	---	ORR and TTNT; 2 years PFS and OS; 2 years DOR

NR: not reported; PFS: progression-free survival; OS: overall survival; EFS: event-free survival; RR: response rate; ORR: overall response rate; PR: partial response and MR: minor response. DOR: duration of response; EDR: early death rate; TTF: time to first treatment; TTNT: time to next treatment; CLL: chronic lymphocytic leukemia and SLL: small lymphocytic lymphoma. Ld: low dose; SEM: standard error of the mean; Int-1: intermediate 1 and MDSs: myelodysplastic syndromes. CMML: chronic myelomonocytic leukemia; NHL: non-Hodgkin’s lymphoma and 13-CRA: 13-cis-retinoic acid.

**Table 2 biomedicines-10-00987-t002:** Relationships between vitamin D levels and main outcomes after allo-HSCT.

	Study	*N*	Vitamin D Levels Mean ± 2 S.D OR Median (Range)		Impact on GvHD	Survival and Other Endpoints
			Pre-Allo	Post-Allo		
**Kreutz et al., 2004 [165]**	Prospective	48, UP	36.4 ± 2.2 nmol/L	↓ compared to pre-allo (27.8 ± 1.3 nmol/L)	In patients with grades 3–4 GvHD, serum levels remained low/dropped (*p* = 0.031)	
**Glotzbecker** **et al., 2013** **[166]**	Retrospective	53, AP	21.9 ng/mL (7.8–45.7); vitamin D cutoff 25 ng/mL		cGVHD at 2 y 63.8% vs. 23.8% (*p* = 0.009); extensive cGVHD at 2 y 54.5% vs. 14.3% (*p* = 0.005)	OS: 53% vs. 50% (*p* = 0.57); PFS: 51% vs. 47% (*p* = 0.61)
**Ganetsky** **et al., 2014 [167]**	Retrospective	54, AP		D + 30 20 ng/mL (6–50)	D30 levels inversely correlate with risk of skin aGvHD in patients undergoing RIC (*p* < 0.001)	
**Campos** **et al., 2014** **[170]**	Prospective	66, PP	25.7 ± 12.3 ng/mL vs. controls 31.9 (*p* = 0.01); deficiency prevalence 32% vs. 8% (*p* = 0.01)	D + 30 22.7 ± 10.7 ng/mL; D + 180 20.9 ± 10.9 ng/mL (*p* = 0.01)	No association with GvHD	No effect on survival
**Beebe** **et al., 2018** **[171]**	Retrospective	72, PP	26 ng/mL (19–34 ng/mL); deficiency 35%	Pre-HSCT and D + 100 similar at 1 year (*p* = 0.01); 35 ± 16 vs. 27 ± 10	No association with GvHD	1-year OS significantly lower among patients with vitamin D deficit (*p* = 0.001)
**Robien,** **et al., 2011** **[172]**	Retrospective	95, PAP		65% had ≥ 75 nmol/L; 24% had low levels (50–75); 11% had < 50 nmol/L	No association with GvHD	
**Urbain,** **et al., 2012** **[161]**	Prospective	102, AP	16.4 ± 8.9 ng/mL; 89.2% had < 30 ng/mL and 23.5% < 10 ng/mL	D + 30 15.5 ± 8.7 ng/mL; D + 100 14.9 ± 7.5 ng/mL	Trend toward higher risk of grade 2–4 aGvHD among patients with lower vitamin D levels (*p* = 0.066)	
**Gjærde,** **et al.,** **2021** **[181]**	Retrospective	116, AP	64 nmol/L; 29% had < 50 nmol/L and 8% had < 25 nmol/L		Pre-HSCT > 85 nmol/L had 1.5 times higher odds of grade II–IV aGvHD than < 47 nmol/L (CI: 0.84–2.7)	
**Bajwa** **et al., 2021** **[173]**	Retrospective	233, PP	24.24 ng/mL All patients had vitamin D insufficiency	D + 30 24.76 ng/mL vs. D + 100 29.89 ng/mL All normal thereafter	No statistical difference in acute or chronic GvHD	No significant influence on OS
**Hansson** **et al., 2014** **[168]**	Retrospective	123, PP	Insufficient-level group (33 nmol/L; 13–49); sufficient level group (63 nmol/L; 50–97)	Vitamin D at 6 months 23 nmol/L (18–24) in moderate/severe cGvHD vs. 37 nmol/L (10–80) in no cGVHD (*p* = 0.004)	Grades 2–4 aGvHD 47% in low vitamin D levels vs. 30% in sufficient (*p* = 0.05)	OS: 87% vs. 50% (*p* = 0.01) for insufficient vs. sufficient level; relapse for insufficient vs. sufficient level groups: 33% vs. 4% (*p* = 0.03)
**Wallace** **et al.,** **2015** **[162]**	Prospective	134, PP	70% insufficient levels (<30 ng/mL); 33% deficient levels (<20 ng/mL)	68% D + 100 insufficient (<30 ng/mL); 31% deficient (<20 ng/mL).	No significant impact on acute or chronic GvHD (*p* = 0.8).	Vitamin D < 20 ng/mL at D + 100 was associated with ↓ OS (70% vs. 84.1%; *p* = 0.044); no impact pre-allo
**Von Bahr** **et al., 2015** **[169]**	Retrospective	166, AP	42 nmol/L (10–118; 53% insufficient levels; 11% deficient); healthy controls (66.5 nmol/L; 21–104; *p* < 0.001)	39 nmol/L (10–116) at 6 months.	No significant impact on aGvHD; in 2-year cGvHD (moderate/severe), deficient vit. D level 56%, insufficient vit. D level 31% and sufficient vit. D level 21% (*p* = 0.01)	2-y OS according to vit. D levels; 63% deficient, 69% insufficient, 76% normal; *p* = 0.24; aa *p* = 0.02; Significant ↑ in CMV disease if deficient vit. D (*p* = 0.005) and ↑ in antibiotics (*p* = 0.011)
**Katic** **et al., 2016** **[174]**	Prospective	310, PAP		Only patients with GvHD; 30 ng/mL (22–42); 77.7% had >20 ng/mL and 22.3% had ≤20 ng/mL	No association between vit. D levels and major cGvHD characteristics	↓ OS in patients with vitamin D ≤20 ng/mL vs. >20 ng/mL
**Perera** **et al., 2015** **[176]**	Retrospective	492, UP			No significant differences in acute/chronic GvHD	Higher mortality in vitamin-D-deficient cohort vs. replete cohort (HR of 1.5; CI of 1.1–2.0; *p* = 0.013); no PFS or relapse differences
**Radujkovic** **et al., 2017** **[177]**	Retrospective	492, AP	11.8 ng/mL (4.0–46.3); vitamin D deficiency in training cohort 80%; in validation cohort 87%		No significant impact on cumulative incidences of acute and chronic GvHD	↓ OS in vitamin D deficiency (HR of 1.78; *p* = 0.007) due to higher risk of relapse (HR of 1.96; *p* = 0.006)
**Peter** **et al., 2021** **[178]**	Prospective	143 + 365, AP	All patients tested for 1,25-dihydroxyvitamin-D3 and 25-hydroxyvitamin-D3 from day −16 to −6 before allo-HSCT	25-hydroxyvitamin-D3 showed a steady increase; 1,25-dihydroxyvitamin-D3 peaked around the time of allo-HSCT	No significant association between vitamin D levels and severe GvHD	↓ 25-hydroxyvitamin-D3 during follow-up or ↓ peritransplant 1,25-dihydroxyvitamin-D3 was associated with increased TRM (*p* = 0.002 and *p* = 0.001).

aa: age adjusted. UP: Unspecified Population; PP: Pediatric population; AP: Adult Population; PAP: Pediatric and Adult Population; GvHD: Graft-versus-Host Disease; cGvHD: Chronic Graft-versus-Host Disease; aGvHD: Acute Graft-versus-Host Disease; OS: Overall survival; PFS: Progression Free Survival; D+: Day +; RIC: Reduced Intensity Conditioning; HSCT: Hematopoietic Stem Cell Transplantation; CI: Confidence Interval; CMV: Cytomegalovirus; HR: Hazard Ratio.

**Table 3 biomedicines-10-00987-t003:** Interventional studies evaluating the administration of vitamin D after allogeneic transplantations.

	Study	*N*	Vitamin D2 or D3 and Dose	Vitamin D Levels		Impact on GvHD NRM and Survival	
				Pre-Allo	Post-Allo		
**Wallace** **et al., 2018** **[184]**	Prospective	10, PP	Cholecalciferol: single enteral dose (maximum 600,000 IU) based on weight and pre-transplantation vitamin D levels	Mean pre-transplantation 25-OH vitamin D level 28.9 ± 13.1 ng/mL	All patients achieved a therapeutic vitamin D level (>30 ng/mL) sustained at or above 8 weeks		
**Silva** **et al., 2011** **[185]**	Retrospective	12, AP	Cholecalciferol: 1000 IU per day (orally) plus calcium carbonate (1250 mg; one pill daily) after HSCT for at least 6 months in patients with osteopenia			All patients had active cGvHD; At 6 months after treatment, 5 patients obtained complete response, 6 patients obtained partial response and 1 patient had no response	
**Duncan** **et al., 2011** **[183]**	Prospective	22, PP	Ergocalciferol: 50,000 IU once weekly for 6 weeks	Mean pre-transplantation 22.8 ng/mL (7–42.6); vitamin D deficiency 37.3% (CI of 25.8–50%).	Mean increase following supplementation 18.8 (SD = 11.3; 8–42); 4.5% remained deficient		
**Bhandari** **et al., 2021** **[186]**	Prospective/historical cohort comparison	33, PP	Cholecalciferol: one-time oral Stoss * dose of cholecalciferol in 5000 IU/mL liquid formulation, 5000 IU/capsule or 50,000 IU/capsule vs. standard dose 14 days before conditioning)	Mean pre-transplantation 27.7 ng/mL (SD 10.8); 59% were vitamin D insufficient vs. 61% in the historical cohort	* Mean level (*p* < 0.001) post Stoss of 72.2 ng/mL vs. standard dose of 35.8 ng/mL; * 97% of Stoss cohort vs. 67% of standard-dose cohort vitamin D sufficient	No association with acute GvHD, veno-occlusive disease or transplant-associated thrombotic microangiopathy	
**Wallace** **et al., 2016** **[187]**	Prospective	60, PP	Cholecalciferol. * control cohort (1) treated according NKF ^ guidelines; * intervention cohort (2): high doses of vitamin D based on body weight (15,000–100,000 IU weekly)	51% (18 of 35 patients) in control cohort and 48% (12 of 25 patients) in the intervention cohort were vitamin-D-insufficient at the time of transplant	Outcomes improved in Cohort 2 but only 64% achieved therapeutic level despite receiving > 200 IU/kg/day		
**Kenny** **et al., 2019** **[156]**	Prospective	144, AP	Cholecalciferol: The dose was guided by vitamin D levels; max. 50,000 IU orally once weekly)	72.9% vitamin-D-deficient before HSCTs; mean pre-transplantation 21 ng/mL	26.4% were vitamin D deficient before HSCTs; mean 6 month post-transplant level 36 ng/mL After treatment, significant difference between Vit D levels pretrans-plant vs. posttransplant (*p* < 0.001)		
**Caballero-Velázquez** **et al., 2016** **[188]**	Prospective	150, AP	Cholecalciferol in three groups: control (CG; no vitamin), low-dose (LdD; 1000 IU/day) and high-dose (HdD; 5000 IU/day)	Plasma levels of 25-OH vitamin D3 were measured on days −5, +1, +7, +14 and +21	Significantly higher levels among patients receiving high doses compared with the control group beyond day +7	↓ overall and moderate + severe cGvHD at 1 year (LdD 37.5% and 19.5% and HdD 42.4% and 27% compared with CG 67.5% and 44.7%; *p* < 0.05); in multivariable analysis, vitamin D ↓ the risks of overall cGvHD and moderate and severe cGvHD (*p* ≤ 0.01); similar relapse and survival rates	
**Carrillo-Cruz** **et al., 2019** **[189]**	Prospective	107, AP	Cholecalciferol in three groups: D3 control (CG; no vitamin D), low-dose (LdD; 1000 IU/day) and high-dose (HdD; 5000 IU/day			Incidences of overall cGvHD varied depending on the VDR genotype among patients with FokI CT genotype, (22.5% vs. 80%; *p* = 0.0004 and among patients treated with vitamin D compared with CG (HR of 0.143; *p* < 0.001) and patients w/o BsmI/ApaI/TaqI ATC haplotype (22.2% vs. 68.8%; *p* = 0.0005)	
**Bhandari, 2020** **[175]**	Prospective	314, PP	Cholecalciferol	Obtained in 94 patients; mean levels of vitamin D with supplementation 33.67 ng/mL vs. 29.16 ng/mL without supplementation (*p* = 0.11)	31.85 ng/mL in patients with aGvHD vs. 31.42 ng/mL in those w/o aGvHD (*p* = 0.91)	Vitamin D levels correlated with OS; for every 10 ng/mL increase, there was a 28% decreased risk of death (*p* = 0.01), but no difference for levels before HSCT; malignant diagnoses were associated (multivar. analysis) with EFS (*p* < 0.01).	

PP: Pediatric population; AP: Adult Population; PAP: Pediatric and Adult Population; HSCT: hematopoietic stem-cell transplantation; GvHD: graft-versus-host disease; cGvHD: chronic graft-versus-host disease; aGvHD: acute graft-versus-host disease; CI: confidence interval; SD: standard deviation; VOD: veno-occlusive disease; TA-TMA: transplant-associated thrombotic microangiopathy; HR: hazard ratio; VDR: vitamin D receptor; SNPs: single nucleotide polymorphisms and OS: overall survival. (*) The Stoss dosing was based on weights and total 25-OHD levels, as previously published by Wallace et al.—vitamin D < 10 ng/mL: 14,000 IU/kg/dose; vitamin D 10–29 ng/mL: 12,000 IU/kg/dose; vitamin D 30–50 ng/mL: 7000 IU/kg/dose; (^) National Kidney Foundation; aggressive dosage increases in those who remained vitamin D insufficient.

**Table 4 biomedicines-10-00987-t004:** Clinical trials currently active and evaluating the use of vitamin D among patients undergoing allo-HSCT.

	*N*	Vitamin D	Dose	Main Objective
**Cincinnati Children’s Hospital Medical Center, 2018**	100		Single large dose of vitamin D “Stoss therapy” with a placebo vs. single large doses of both vitamins D and A	Investigate incidences of acute GI GvHD at day +100 after transplant
**Cincinnati Children’s Hospital Medical Center, 2021**	20	Cholecalciferol	Vitamin D OTF weekly for a maximum of 12 weeks. The dose may be increased or decreased based on the dosing schema	Investigate efficacy of OTF D3 replacements by measuring vitamin D levels
**Children’s Hospital Los Angeles, 2018**	33	Cholecalciferol	Single ultra-high dose of vitamin D	Investigate incidences of GvHD, veno-occlusive disease and thrombotic microangiopathy at day +100 after transplant
**Cincinnati Children’s Hospital Medical Center, 2016**	10	Cholecalciferol	One oral vitamin D dose (based on vitamin D status and rounded to 5000 IU) <2 weeks prior to HSCT	Investigate vitamin D sufficiency following Stoss dosing prior to transplant
**University of British Columbia, 2018**	84	Cholecalciferol	Intervention group: loading dose of 100,000 IU of vitamin D3 after 2000 IU of vitamin D3 daily.	Test efficacy and safety of high-dose vitamin D therapy by measuring serum 25-OH vitamin D levels weekly for 8 weeks
**Seoul National University Hospital, 2017**	88	Cholecalciferol	Control group: 2000 IU vitamin D3 daily	Assess efficacy (in patients achieving sufficient serum 25-OH vitamin D3 levels on day +100 post-aHSCT) of 100.000 IU of vitamin D3 prior to aHSCT

GI: gastrointestinal; GvHD: graft-versus-host disease; OTF: oral thin film; HSCT: hematopoietic stem-cell transplantation; aHSCT: allogenic hematopoietic stem-cell transplantation and cGvHD: chronic graft-versus-host disease.

## Data Availability

This is a biblografic review article, no original data was originated in this work.

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
