# Peer review of "Effect of Vitamin D on Graft-versus-Host Disease"

_biomedicines, 2022, doi:10.3390/biomedicines10050987_

Round 1

Reviewer 1 Report

The review provides a nice review of the impact of Vitamin D on the immune system with a particular focus on the immunobiology of hematopoietic cell transplantation. The authors provide comprehensive and useful tables which summarize the work that has been done in the field over the past years. One major criticism is that this review does not provide enough information on important hematopoietic cell transplantation variables (such as stem cell source, patient age, transplant platform).

Major Points:

  • Authors should provide insights into the differences in Vitamin D levels between young vs adult individuals and comment on the potential differential impact on cells on the immune system. Information on the age of the cohorts reported in Table 2 and 3 (pediatric vs adult) should be included.
  • Table 2 and 3 should be expanded including important variables that could impact on the risk of GVHD such as stem cell source (PBMC, cord blood, BM) and transplant platforms (sibling, MUD, haplo).

Minor Points:

  • Sometimes, Vitamin D is mentioned in the text as VD. We would suggest the use of Vitamin D and not VD throughout the manuscript
  • line 91 dot is missing
  • line 107 - dot is missing
  • line 190 - dot
  • line 216- dot is missing
  • ATRA is first mentioned in line 274 directly as an abbreviation
  • line 279 - point before and after []
  • line 282 - point before and after []
  • line 391 - point before and after []
  • line 422- is missing 0 before .006
  • line 475- is missing 0 before .013
  • line 477- is missing 0 before .055
  • DFS is first mentioned in line 501 directly as an abbreviation

Reviewer 2 Report

to be published

Reviewer 3 Report

The present manuscript provides a detailed overview of the association between vitamin D (intake/concentrations) and the immune system, focusing on allogeneic  hematopoietic stem cell transplantation and on treatment of hematologic malignancies. It is obvious that the authors have done their best to identify the important papers and to present these in a general overview.

The manuscript is well written and deserves the attention of the reader. This reviewer would like to read the opinion of the authors on the various topics raised in the manuscript as based on the various outcomes as presented in the various papers used. An overall synthesis of the findings would be appreciated.

Despite the effort by the authors there are some issues to be mentioned/discussed in this review:

  • Vitamin D as causative, secondary factor or as biomarker of an associated primary factor. In the manuscript there should be some room for a proposed causation of effect. Do the authors have any reason to believe that vitamin D concentration or intake directly affects the immune system giving rise to the various conditions or does the molecule act as an intermediate factor. Is deficiency in 25(OH)D, calcitriol, etc. not a marker of a relatively low immune status? The authors should discuss their thoughts on this.
  • Graft versus host and hematological diseases are often age-related, occurring in relatively higher numbers in older persons than in younger. To which extent do the authors believe that aging is more of importance on the immunological issues while vitamin D is a mere cofactor?
  • Especially in Table 3 the intervention arm consists of many multifactorial experimental factors. This makes the appreciation of a potential effect by vitamin D (levels) difficult to assess. Please try to acknowledge this fact in the text more clearly. In the end an association between 25(OH)D levels and the primary outcome is more of importance acknowledging the many other factors having their potential effect.
  • Confounding factors, such as baseline levels of 25(OH)D might have a strong impact on the outcome of the respective studies. Can the authors throw any light on this. Is this issue presented in the many publications as listed in the current manuscript?
  • Power of the various studies should be mentioned in the various overviews. It is clear that the power between the various studies is extremely different and therefore also the conclusions to be reached from these studies. Do the authors believe that under powering might have influenced the overall conclusions of the present study.
  • In Table 2: please provide p-values with respect to the GHVD in order to get more grip on the power of the study. The same holds true for Table 3: Vitamin D levels pre/post and Impact on GVHD NRM and Survival. Please specify the power of the studies listed in Table 4 so that the reader can appreciate the potential relevance of the respective studies.
  • Why didn’t the authors perform a meta-analysis of the various studies in which power is one of the important determinants. This reviewer would favour such an approach, especially due to the more unbiased impact on the outcome.
  • Unfortunately there is no attention in the manuscript on the modelling work of vitamin D intake/levels on the various aims of the present manuscript. Is there any evidence that modelling papers strengthen the conclusions reached by the authors?
  • Did the human dosing studies clearly demonstrate a dose-dependent effect on the main aim of their study, see e.g. Table 2?
  • To which extent do the authors believe that there is a limit in the dose to be given to humans? Is there any evidence of side effects by a too high dosing. There are suggestions of an U-shaped relationship between 25(OH)D levels and the incidence of various types of cancer (Michaëlsson, K. et al. Am. J. Clin. Nutr. 2010, 92, 841–848; Helzlsouer, K.J. Cancer Prev. Res. 2010, 3, 4–5).
  • Is there any substantiation of the results of in vitro research by human data? Are these (cellular) models reliable enough? The same question could be asked for animal data. In other words can the authors advise on the most reliable/standardized models to be used before going to human studies?
  • Could the authors discuss the potential impact by obesity on the application of vitamin D supplementation due to the expected fat storage of (supplemented) vitamin D.
  • Please provide some information on the uptake mechanisms of the various forms of vitamin D, such as calcitriol, ergocalciferol, etc. Is there some effect by the specific molecule on biological activity, such as presented for ergocalciferol versus cholecalciferol. Moreover, could “pharmacology” explain some of the results as observed in the various studies presented?
  • This reviewer would appreciate an overall hypothesis about the primary aim at the end of the manuscript, putting the observations into perspective.

Minor:

  • Please write in vivo and in vitro in italics.
  • Line 231: proposes.
  • Line 232: defines.
  • Table 2: median +/- 2 x SD values are provided. Please be aware that median values describe a parameter free distribution while mean values a parametric. Please use median and range versus mean and 2 x +/- SD when appropriate.

Reviewer 4 Report

The review article “Effect of vitamin D on graft -versus-host-disease” summarize the current state of understanding on the topic. The authors performed a systematic literature search using 176 articles. The authors presented all important timelines in vitamin D major events such metabolism, classical and non-classical effects, clinical settings with special data in hematological malignancies and stem cell transplantation. There is a high quality of discussion regarding the active role of vitamin D on GVHD pathogenesis. Emphasizing the finding of a decreased risk of cGvHD among specific SNP, the effect of vitamin D in dendritic cells population might be the most relevant to justify the impact of vitamin D on cGvHD incidence, and  support its use in allo stem cell transplantation setting. The review article contains a large amount of detailed information, many clinical and preclinical studies, but its structure and flow are very easy to follow. The summary presents an unbiased information of the current understanding of the topic. 

Reviewer 5 Report

See the file attached

Round 2

Reviewer 3 Report

This reviewer thanks the authors for their reaction on the various issues raised during the first review procedure. Unfortunately they do not provide all the information as requested. The present reviewer would have appreciated to get their opinion in the manuscript itself, instead of trying to answer the question only for the reviewers.

Unfortunately I cannot find the revised manuscript, therefore the reaction is purely on the answers as provided by the authors on the issues raised.

The concerns of the present reviewer remain the absence of a discussion on power and thereby on the relevance of the outcome of the various studies reviewed by the authors. When performing a meta-analysis this aspect would become much clearer, but unfortunately was not the aim of the present overview. Moreover, when considering the studies as listed in the manuscript the importance of many confounding factors on the primary aim might lead to a substantial rise of various aspects of bias. Finally the inclusion of the author’s opinion with respect to their view on a direct versus an indirect impact by vitamin D on the graft-versus-host phenomena would be greatly appreciated. This reviewer had hoped that by adding some more details of the above more information and in-depth knowledge could be gained by the reader. This would enhance the message of the authors profoundly, at least to the opinion of the present reviewer.

Some more specific issues:

  • Vitamin D as causative vs intermediate parameter in the physiological context of the paper.

This reviewer truly believes that the authors should emphasize this difference with respect to a direct vs. an indirect impact by vitamin D more clearly in the text. The positive effect on hematopoietic conditions after supplementation of vitamin D is contradicted by studies showing no effect. Moreover, supplementation itself should be addressed in a dose-dependent fashion while standardizing the various confounding factors. In itself, even when vitamin D is considered to be an intermediate factor, the observed positive findings deserve further attention. Unfortunately the reviewer cannot find the alterations in the text at lines 478-491 addressing the above as explained above.

  • Impact by confounding factors, such as aging, on the outcome.

This reviewer cannot check the alterations by the authors, but it is of eminent importance to stress this fact to the reader.

  • Impact by baseline levels on the outcome.

This reviewer understands that there is no specific mentioning of this potential effect. Even when in a certain target population only low levels of 25(OH)D at the start is being studied, this factor might/will still have a substantial impact on the primary aim.

  • Power of the used studies.

Thank you for your statement in the text as mentioned in your reaction.

  • P-values reported.

Thank you for the inclusion of p-values when provided by the various manuscripts as used in the current analysis. Unfortunately not all have been reported making it difficult to assess an indication of the power of the various studies as applied in the manuscript.

  • Dose-dependent effect of vitamin D.

This reviewer fails to understand the logic of the authors’ reaction on a potential dose-dependent effect of vitamin D administration/supplementation, by stating that many retrospective studies have been conducted and that the prospective studies are in line with the outcome of the former. Please acknowledge at least the various confounding factors beside the fact that no attempt was made to analyze any true dose/concentration association between vitamin D intake/concentration and outcome specifically.

The present reviewer believes that the authors have thrown a very important light on the association between vitamin D and graft-versus-host conditions. Their manuscript gives detailed information on various physiologically important interactions, enabling the reader to appreciate a difficult biomedical topic more clearly. As stated earlier the manuscript is well written and provides a rapid introduction on the above mentioned association. Therefore, despite the various remaining issues this manuscript deserves publication.